# Retatrutide—A Game Changer in Obesity Pharmacotherapy

**DOI:** 10.3390/biom15060796

**Published:** 2025-05-30

**Authors:** Vasiliki Katsi, Georgios Koutsopoulos, Christos Fragoulis, Kyriakos Dimitriadis, Konstantinos Tsioufis

**Affiliations:** 1st Cardiology Clinic, National & Kapodistrian University of Athens, 11527 Athens, Greece; giorgoskoutsopoulos93@gmail.com (G.K.); christosfragoulis@yahoo.com (C.F.); dimitriadiskyr@yahoo.gr (K.D.); ktsioufis@gmail.com (K.T.)

**Keywords:** retatrutide, obesity, incretins, type 2 diabetes mellitus

## Abstract

Obesity and type 2 diabetes mellitus (T2DM) are global health crises with significant morbidity and mortality. Retatrutide, a novel triple receptor agonist targeting glucagon-like peptide-1 (GLP-1), Glucose-Dependent Insulinotropic Polypeptide (GIP), and glucagon receptors, represents a groundbreaking advancement in obesity and T2DM pharmacotherapy. This review synthesizes findings from preclinical and clinical studies, highlighting retatrutide’s mechanisms, efficacy, and safety profile. Retatrutide’s unique molecular structure enables potent activation of GLP-1, GIP, and glucagon receptors, leading to significant weight reduction, improved glycemic control, and favorable metabolic outcomes. Animal studies demonstrate retatrutide’s ability to delay gastric emptying, reduce food intake, and promote weight loss, with superior efficacy compared to other incretin-based therapies. Phase I and II clinical trials corroborate these findings, showing dose-dependent weight loss, reductions in Glycated Hemoglobin (HbA1c) levels, and improvements in liver steatosis and diabetic kidney disease. Common adverse effects are primarily gastrointestinal and dose-related. Ongoing Phase III trials, such as the TRIUMPH studies, aim to further evaluate retatrutide’s long-term safety and efficacy in diverse patient populations. While retatrutide shows immense promise, considerations regarding cost and the quality of weight loss beyond BMI reduction warrant further investigation. Retatrutide heralds a new era in obesity and T2DM treatment, offering hope for improved patient outcomes.

## 1. Introduction 

Obesity has emerged as a pressing global health crisis, often referred to as a pandemic due to its widespread prevalence and significant impact on public health. It is estimated to be responsible for approximately 3.4 million deaths each year worldwide, underscoring its severity as a major contributor to mortality rates [1]. Clinically, it is characterized by an excess accumulation of adipose tissue within the body, and it is formally defined as a body mass index (BMI) of ≥30 kg/m^2^. It can affect patients both on a psychological and physical level, with overweight and obese people found to be at risk for diabetes, cardiovascular disease, cancer, and premature death [2]. This has a significant toll on life expectancy, which is shown to be reduced by 7 years at the age of 40 years [3]. Added to that, it is constantly growing in scale with an estimated projection of overweight and obese adults at 1.35 billion and 573 million, respectively, by the year 2030 [4].

Coming hand in hand with obesity, type 2 diabetes mellitus (T2DM) poses a considerable threat to individual health and a significant burden on the world’s financial health policy. Described as a disorder of glucose homeostasis and insulin sensitivity, it is primarily triggered by uncontrolled hyperglycemia, and it is responsible for a large number of microvascular (like diabetic retinopathy, diabetic nephropathy, and diabetic neuropathy) and macrovascular complications (like diabetic foot syndrome, cerebrovascular events, and myocardial infarction) [5]. The presence of T2DM also increases susceptibility to a variety of cancers (such as liver, biliary tract, pancreas, stomach, colorectal, kidney, bladder, breast, and endometrial) and infections (like respiratory, urinary tract, and skin and soft tissue infections) [6].

A tight interplay between these two is a common finding in many patients, and many metabolic pathways show their close pathophysiological connection. For example, the global prevalence of metabolic dysfunction-associated steatotic liver disease (MASLD) and metabolic dysfunction-associated steatohepatitis (MASH) among diabetic patients is estimated at 65.33% and 66.44%, respectively [7].

Unhealthy diets and sedentary lifestyles are pivotal predisposing factors contributing to the growing global burden of obesity, T2DM, and metabolic dysfunction-associated fatty liver disease (MAFLD). Diets rich in saturated fats, refined carbohydrates, and ultra-processed foods—hallmarks of the Western dietary pattern—have been consistently linked to excessive adiposity, insulin resistance, and chronic systemic inflammation [8]. These conditions are central to the pathogenesis of both T2DM and MAFLD, forming a critical component of the broader metabolic syndrome. Additionally, prolonged physical inactivity further compounds these risks by diminishing metabolic flexibility and impairing insulin sensitivity, independent of body weight [9]. The synergistic effect of poor dietary quality and lack of exercise creates a fertile ground for the development and progression of metabolic disorders.

Intervention through lifestyle modification remains the first-line therapeutic strategy. Notably, adherence to the Mediterranean diet—a dietary pattern rich in fruits, vegetables, whole grains, olive oil, and lean proteins—has demonstrated significant improvements in hepatic steatosis and insulin resistance among patients with MAFLD [10]. Furthermore, structured exercise regimens incorporating both aerobic and resistance training have been shown to reduce liver fat content and improve glycemic control [11]. However, despite the well-documented benefits of lifestyle changes, maintaining long-term adherence remains a challenge for many individuals, often requiring multidisciplinary support and long-term behaviour change strategies. When necessary, pharmacological interventions are essential to eliminate the gap for effective prevention and management of these interrelated conditions.

Numerous pharmacological treatments seek to disrupt these harmful connections, with a considerable number focusing on the incretin hormonal pathways and appetite regulation. Three of the most common incretin peptide hormones are Glucagon-Like Peptide (GLP-1), Glucose-Dependent Insulinotropic Polypeptide (GIP), and Glucagon. The emergence of GLP-1 receptor agonists, such as the first clinically approved agonist, exenatide, in 2005, paved the way for the thorough exploration of these molecules [12]. Shortly, other molecules such as liraglutide, dulaglutide, and semaglutide came forth, offering a once-weekly dosage. Beyond achieving more effective glucose control and offering worthy weight loss, these medications achieved a reduction in rates of major adverse cardiovascular effects, heart failure, kidney disease, and cardiovascular death [13,14,15]. These promising results led to the development of the first dual agonists, with tirzepatide, a GIP and GLP-1 receptor agonist, being the most prominent one. Tirzepatide achieved significant reductions in Glycated Hemoglobin A1c (HbA1c) levels and body weight compared to placebo. Additionally, tirzepatide was found to have a more pronounced effect compared with the GLP-1 agonist semaglutide in people with T2DM. The incidence of gastrointestinal (GI) adverse events was increased compared with placebo; however, neither tirzepatide nor semaglutide increased the risk of serious adverse events or severe hypoglycaemia [16].

A first-of-its-class triple receptor agonist, retatrutide, aims to deliver a revolution in obesity and T2DM pharmacotherapy. This groundbreaking medication is shown to enhance therapeutic outcomes by delivering more pronounced reductions in both body weight and HbA1c levels than current pharmacotherapies.

## 2. Materials and Methods

A review of the literature was conducted to examine the potential impact of the emerging drug retatrutide on T2DM and obesity pharmacotherapy. The search was carried out on Medline using the keywords: retatrutide, T2DM, obesity, and novel pharmacotherapy, in combination with the boolean operators AND or OR. Only articles published in English from 2005 onwards were included, with the most recent search conducted in April 2025.

## 3. Targeted Molecules

### 3.1. Glucagon-like Peptide (GLP-1)

GLP-1 is an incretin hormone that is predominantly secreted from the intestinal tract, mainly as a response to meals. Small Intestine L-cells are responsible for their production from a proglucagon gene [17]. Production of this molecule is also shown to occur in pancreatic α-cells and the central nervous system in areas like the nucleus tractus solitarius and microglia [18]. It predominantly acts by activating a receptor that induces stimulation of insulin release, decreases glucagon secretion, reduces food intake, and delays gastric emptying (GE) by affecting gastric and intestinal motility [19]. GLP-1 receptors can be found in a variety of tissues such as the pancreas, gastric mucosa, kidney, lung, heart, skin, immune cells, and brain. They are 463 amino acid heptahelical G protein-coupled receptors that, when activated, stimulate cyclic adenosine monophosphate (cAMP) formation followed by activation of protein kinase A pathways [20].

Especially regarding the central nervous system, the expression of GLP-1 receptors in reward-related regions, such as the hypothalamus, amygdala, nucleus accumbens, paraventricular nucleus, ventral tegmental area, locus coeruleus, and brain stem [21] may suggest a possible association between reward learning and symptoms like anhedonia in diseases such as depression. However, the multifactorial and polygenic nature of this diagnosis requires further investigation before establishing a rigid conclusion [22]. GLP-1 activation is also shown to reduce inflammation by reducing macrophage activity [23]. Its effects on satiety and food intake are found in healthy individuals as well as in obese and diabetic patients [24].

### 3.2. Glucose-Dependent Insulinotropic Polypeptide (GIP)

Also excreted from intestinal K cells, GIP is a protein mostly postprandially secreted [25]. It consists of 42 amino acids that take form after proteolysis of a 153-amino acid precursor. GIP receptors exist as two isoforms, consisting of 466 and 493 amino acids, respectively, that, when activated, display an increase in intracellular calcium and arachidonic acid. Their activation is also related to adenylyl cyclase activation. By acting on these receptors that are expressed in multiple tissues, including primarily pancreatic β-cells and secondarily adipose and nervous tissue, GIP stimulates glycogen secretion both in normoglycemic and hyperglycemic states. It also reduces appetite by having a negative impact on GE and by enabling insulin release from the pancreas, acting in tandem with GLP-1 [26]. However, it acts in contrast with GLP-1 as much as glucagon release is concerned by enhancing its release. Furthermore, growth factor-dependent pathways such as MAPK (extracellular signal-regulated kinases 1 and 2 [ERK 1/2]) and protein kinase B are found to be affected by GIP activation [20]. GIP effects regarding adipose tissue are also described as it is shown to stimulate adipogenesis, inhibit lipolysis, and stimulate de novo lipogenesis [27]. Additionally, GIP is shown to have a direct influence on bone, inducing osteoblast activity [28].

### 3.3. Glucagon

Finally, glucagon, formed from a precursor pro-glucagon molecule of 180 amino acids, is a hormone that is mostly found in pancreatic α-cells [19]. Production in the small intestine is also described [28]. After its discovery in 1921, its contribution to the pathophysiology of type 2 diabetes mellitus opened a new age in antidiabetic medications and in the study of metabolic diseases. It consists of 29 amino acids [29] and its gene is located on chromosome 2 [30]. Glucagon receptors are mainly found in the liver hepatocytes and in the kidneys. Less expression is also described in other tissues such as the heart, pancreas, adipose tissue, and the GI tract. Glucagon receptors are G protein-coupled receptors that, when stimulated, lead to an increase in intracellular cAMP and calcium that, in advance, activate the protein kinase A pathway [31,32]. Its main drive of release is low glucose blood levels, and glucagon receptors are found in many tissues, with hepatocytes being the most common place. Glucagon allows gluconeogenesis and glycogenolysis in the liver while blocking glycogenesis. These effects lead to high blood glucose levels. Glucagon also reduces GI motility [33] and, by acting on adipose tissue, decreases lipogenesis and induces lipolysis, leading to increased production of non-esterified fatty acids and ketone bodies. Glucagon is also found to affect brown fat in animal models, inducing thermogenesis and energy expenditure. However, human experiments have not proven more than a modest effect on energy expenditure, not achieving clinical significance [28].

## 4. General Aspects of Retatrutide

### Molecular Structure—Pharmacology

A promising new therapy that targets all the previously mentioned molecules is the triple receptor agonist retatrutide. Retatrutide is a synthetic peptide acting as an agonist of GLP-1, GIP, and glucagon receptors (Figure 1). Retatrutide’s engineering allows it to bind uniquely to these receptors [34].

Retatrutide develops in a single continuous helical structure that allows it to run through the receptor’s transmembrane domain with its N-terminal segment. The C-terminal segment takes part in interactions with the N-terminal α-helix of the extracellular domain, the extracellular tip of the transmembrane helix 1 of the GLP-1 receptor, and the extracellular loop 1 of the GIP receptor. This molecule is more potent at the human GIP receptor (EC_50_: 0.0643 nM) and less potent at the GLP-1 (EC_50_: 0.775 nM) and glucagon (EC_50_: 5.79 nM) receptors [35]. It has a dose-dependent action and causes a decrease in gastric emptying, while with a half-life of 6 days, it can mostly be used on a weekly basis. Retatrutide shows mostly hepatic metabolism but does not interact with cytochrome P450 enzymes [36].

Its effects lead to significant weight reduction and to a decrease in HbA1c levels [37]. The most commonly occurring adverse effects are nausea, diarrhea, vomiting, and constipation. However, these resulted in not-so-rare incidents of therapy discontinuation, and their intensity was clearly linked with higher dosages. Practical studies indicate that the discontinuation rates of GLP-1RAs can reach between 20% and 50% within the first year, and patients often use significantly lower dosages compared to those tested in clinical trials [38]. Less common adverse effects were a temporary increase in alanine aminotransferase (ALT) levels, increased heart rate, and skin hyperesthesia [39].

## 5. Insights from Animal Studies

In recent years, retatrutide has amassed a respectable number of animal and clinical studies (Table 1).

With delayed GE being an important effect of weight-lowering drugs acting on glucagon, Urva et al. attempted to investigate retatrutide’s effect, compared with other selective GLP-1 receptor antagonists on animal models [40]. C57/B16 male obese mice (Jackson) were singly housed and maintained on a standardized diet (TD95217; Teklad) with ad libitum water; 16 h before the assessment of acute GE, mice were fasted overnight and treated subcutaneously with either vehicle (10 mL/kg; 40 mM Tris pH 8), long-acting glucagon receptor agonist, semaglutide, retatrutide, or combined semaglutide and long-acting glucagon receptor agonist. Mice were administered 0.5 mL of semi-liquid by oral gavage, and GE was subsequently assessed. GE delay, body weight, and food intake were also assessed following chronic (daily for 10 days) treatment with vehicle, semaglutide, long-acting glucagon receptor agonist, retatrutide, or combined semaglutide and long-acting glucagon receptor agonist. Retatrutide achieved delayed GE in mice in a dose-dependent fashion, as well as semaglutide. The dual combination of them did not manage to differ statistically from the usage of retatrutide alone. 10 nmol/kg of retatrutide achieved a bigger weight drop over 10 days than the other combinations. Furthermore, retatrutide showed the largest decreases in food intake, and chronic treatment for 10 days attenuated the described effects. However, there are some limitations to the study. The method used in the study is not the gold standard scintigraphy that involves radioactive tracers, and the researchers did not take into account secretion-based error [40].

Another study by Ma et al. aimed to compare the effects of retatrutide, liraglutide, and tirzepatide on diabetic kidney disease in db/db mice [41]. The seven-week-old male C57BL/KSJ diabetic db/db and db/m mice were housed in an ambient temperature of 21 ± 2 °C and humidity of 45 ± 10%, maintained under a 12 h light/dark cycle. At 8 weeks of age, the mice were randomly divided into 5 groups (each n = 6): db/m, db/db, db/db + Lira, db/db + Tirz, and db/db + Reta. Each treatment group was divided into two cages with three mice in each cage. The mice in db/db + Lira, db/db + Tirz, and db/db + Reta groups received daily subcutaneous injections of 10 nmol/kg of liraglutide, tirzepatide, and retatrutide for 10 weeks, respectively. The db/m and db/db groups were given an equal amount of saline intraperitoneally over the same time. The dose of the drug was adjusted weekly throughout the study period based on changes in the weight of the mice. Weekly, a glucometer was used to assess the fasting blood glucose (FBG) levels. HbA1c values were measured at the beginning and end of the experiment. After a duration of 10 weeks, urine samples were collected from individually housed mice in metabolic cages over a 24 h period, followed by recording of urine volumes and storage at −80 °C for further analysis. Subsequently, the mice were anesthetized, and blood samples were obtained from the intraorbital venous plexus. After the completion of the experiment, a histopathological analysis was performed, followed by immunohistochemical staining. At the end of the experiment, the weight of mice in the db/db + Reta group decreased, and at 10 weeks, db/db + Reta group mice had the lowest levels of body weight and food intake, followed by db/db + Tirz group and db/db + Lira group, with a statistically significant difference. These findings strongly indicate that all three drugs are effective in reducing food intake and promoting weight loss, with retatrutide demonstrating the most significant impact. Regarding HbA1c levels, retatrutide showed lower levels than those in the db/db group. In addition, the db/db + Reta group demonstrated superior efficacy in reducing alanine aminotransferase, aspartate aminotransferase, total cholesterol, triglyceride, and low-density lipoprotein levels. Finally, after immunohistochemical analysis, retatrutide exhibited lower expression levels of inflammatory and fibrotic mediators compared to tirzepatide and liraglutide, highlighting its potential in the treatment of diabetic kidney disease.

Last but not least, Jall et al. showed a reversal of steatohepatitis in female mice by resulting in enhanced glucose tolerance, decreased body weight and fat mass, and better lipid profiles, thereby aiding in the resolution of hepatic steatosis [42]. Both female and male C57BL/6J mice at an eight-week age received a diet rich in fat and sugar. Due to differences in obesity progression between the two sexes, another cohort of male C57BL/6J mice was switched from a regular diet to a high-fat, high-sugar diet at 30 weeks of age. The mice were maintained at a temperature of 23 ± 1 °C, constant humidity, and on a 12 h light-dark cycle. After 38 weeks of age, mice were randomized within the three cohorts and equally distributed according to body composition. Mice were subcutaneously injected with the triple agonist and immediately followed by fasting for a 4 h period. Afterwards, formalin-fixed liver samples were collected. Histological analysis showed that up to 88.9% and 62.5% of female and male vehicle-treated mice, respectively, were diagnosed with definite steatohepatitis. The administration of the triple agonist resulted in the improvement of steatohepatitis in a dose-dependent fashion, with some female mice showing a complete resolution.

## 6. Insights from Human Studies

### 6.1. Phase I Trials

Coksun et al. in Singapore conducted a phase 1 study to assess the safety and the pharmacokinetic profile of retatrutide [35]. 47 healthy individuals entered the study, while 45 of them received at least one dose. Maximum concentration was achieved within 12–72 h after dosing. The mean half-life was about 6 days. When compared with placebo, retatrutide achieved similar changes in baseline fasting glucose levels. After dosing at 4.5 and 6 mg, mean fasting glucagon levels were decreased from 24 h post-dose up to day 15. Retatrutide achieved a decrease in mean body weight at all dose levels except at the dose of 0.1 mg. The decrease in weight was at its peak at dose levels of 3–6 mg. The most common study treatment-related adverse effects were GI disorders such as vomiting, abdominal distention, and nausea, however, mostly of mild intensity.

Following their animal studies, Urva et al. designed a phase 1b, multiple-ascending dose study in type 2 diabetic patients [40]. These patients received retatrutide once weekly (in dosage groups of 0.5, 1.5, 3, 3/6, and 3/6/9/12 mg), placebo, or 1.5 mg of dulaglutide. Retatrutide 0.5, 1.5, and 3 mg were administered for 12 weeks. The 3/6 mg group received 3 mg of retatrutide for weeks 1–4 and 6 mg for weeks 5–12. Finally, the 3/6/9/12 mg group received 3 mg for the first two weeks, 6 mg for weeks 3–4, 9 mg for weeks 5–8, and 12 mg for weeks 9–12. Gastric emptying was assessed 2 days pre-treatment and 24 h after the dose at day 2, day 30, and day 79. In summary, 72 of the participants received at least 1 dose of the drug, while 43 completed the study. The mean study age was 58.4 ± 7.4 years, mean ΒΜΙ was 32.1 ± 5.1 kg/m^2^, HbA1c 8.66 ± 0.92% [43]. The greatest effect on GE was found after the first retatrutide dose, while it was lower on subsequent doses, despite up-titration.

### 6.2. Phase II Trials

In a phase II double-blind, randomized, placebo-controlled trial, Jastreboff et al. administered retatrutide in doses up to 8 mg for a total of 48 weeks in patients with a BMI ≥ 30 kg/m^2^ or with a BMI of 27–30 kg/m^2^ and a coexisting weight-related condition [39]. The study’s primary endpoint was the percentage change in body weight from baseline to 24 weeks of therapy. The secondary endpoints were the percentage change in body weight from baseline to 48 weeks and the presence of weight reduction in ≥5%, ≥10%, and ≥15%. 338 adults were enrolled in total. The least-squares mean percentage change in body weight at 24 weeks in the retatrutide groups was −7.2% in the 1 mg group, and up to −17.5% in the 12 mg group, as compared with −1.6% in the placebo group. Additionally, at 48 weeks, the same percentages were −8.7% in the 1 mg group, −24.2% in the 12 mg group, and −2.1% in the placebo group. At 48 weeks, a weight reduction of ≥5%, ≥10%, and ≥15% had occurred in 100%, 93%, and 83% of those who received 12 mg of retatrutide and 27%, 9%, and 2% of those who received placebo. The most common adverse effects were also gastrointestinal and dose-related and were partially mitigated with a lower starting dose (2 mg vs. 4 mg). Therefore, in adults with obesity, retatrutide treatment for 48 weeks resulted in substantial reductions in body weight.

In a similar manner, Rosenstock et al. [37] conducted a randomized, double-blind, phase 2 trial where adults aged 18–75 years with type 2 diabetes, HbA1c of 7.0–10.5%, and BMI of 25–50 kg/m^2^ were eligible for enrolment. Participants were treated with diet and exercise alone or with a stable dose of metformin for at least 3 months before screening. They were also randomly assigned to receive once-weekly injections of placebo, 1.5 mg dulaglutide, or retatrutide maintenance doses of 0.5 mg, 4 mg (starting dose 2 mg), 4 mg (no escalation), 8 mg (starting dose 2 mg), 8 mg (starting dose 4 mg), or 12 mg (starting dose 2 mg). The primary endpoint was a change in HbA1c from baseline to 24 weeks, and secondary endpoints included a change in HbA1c and body weight at 36 weeks. At 24 weeks, least-squares mean changes from baseline in HbA1c with retatrutide were –0.43% for the 0.5 mg group, –1.39% for the 4 mg escalation group, –1.30% for the 4 mg group, –1.99% for the 8 mg slow escalation group, –1.88% for the 8 mg fast escalation group, and –2.02% for the 12 mg escalation group, versus –0.01% for the placebo group and –1.41% for the 1.5 mg dulaglutide group. HbA1c reductions with retatrutide were significantly greater (*p* < 0.0001) than placebo in all but the 0.5 mg group and greater than 1.5 mg dulaglutide in the 8 mg slow escalation group (*p* = 0.0019) and 12 mg escalation group (*p* = 0.0002). Findings continued at 36 weeks. Bodyweight decreased dose-dependently with retatrutide at 36 weeks by up to 16.94% for the 12 mg escalation group, compared to 3.00% with placebo and 2.02% with 1.5 mg of dulaglutide. It was evident that retatrutide showed clinically meaningful improvements in glycaemic control and impressive reductions in body weight in type 2 diabetic patients while being relatively safe.

Another phase II trial by Sanyal et al. [44] attempted to show retatrutide’s effects on liver steatosis. 98 patients having ≥10% liver fat content (as defined by magnetic resonance imaging proton density fat fraction) were randomly assigned to 48 weeks of once-weekly subcutaneous retatrutide (1, 4, 8, or 12 mg dose) or placebo. In total, 76 participants managed to complete this substudy. The mean relative change from baseline in liver fat at 24 weeks was −42.9% (1 mg), −57.0% (4 mg), −81.4% (8 mg), −82.4% (12 mg), and +0.3% (placebo). At 24 weeks, normal liver fat (<5%) was achieved by 27% (1 mg), 52% (4 mg), 79% (8 mg), 86% (12 mg), and 0% (placebo) of participants. At 48 weeks, the mean relative change from baseline in liver fat was −51.3% (1 mg), −59.0% (4 mg), −81.7% (8 mg), −86.0% (12 mg), and −4.6% (placebo). Additionally, total liver fat content < 5% at 48 weeks was achieved with 8 mg (in 89% of participants) and 12 mg doses (in 93% of participants). Liver fat reduction was associated with weight loss, abdominal fat, and improved insulin sensitivity and lipid metabolism.

## 7. The Future—Phase III Trials

The promising results of the above trials led to the design of the first phase III trials for retatrutide—the TRIUMPH trials, where the efficacy and safety of retatrutide in participants who are obese or overweight will be evaluated. The first trial—TRIUMPH 1 is estimated to last until May 2026, and it aims to enrol 2300 patients [45].

The TRIUMPH-2 trial will evaluate the efficacy and safety of retatrutide in participants with T2DM who are obese or overweight, including a subset of participants who have obstructive sleep apnea (OSA). It aims to enrol 1000 patients, and it will last about 89 weeks with an expected due date of May 2026.

Another Phase III Trial—TRIUMPH 3 aims to investigate the per cent change from baseline in body weight in obese individuals with established cardiovascular disease (i.e., prior myocardial infarction). Participants must have excess weight, a history of heart attack, stroke, or presence of peripheral arterial disease, and also a history of at least 1 self-reported unsuccessful dietary effort to reduce body weight in order to meet the eligibility criteria. Patients will be excluded (1) if they have had a heart attack, stroke, hospitalization for congestive heart failure or unstable angina within 90 days before screening, (2) if they have been taking any weight loss drugs, including over-the-counter medications, within 90 days before screening, (3) if they have had a change in body weight greater than 11 pounds within 90 days, (4) if they have or are planning a surgical treatment for excess weight, (5) if they have Type 1 Diabetes Mellitus, (6) if they have a family or personal history of medullary thyroid carcinoma (MTC) or multiple endocrine neoplasia syndrome type 2 (MEN-2) and (7) if they have had pancreatitis. The trial is expected to be completed in February 2026 with an enrolment goal of 1800 patients.

Finally, the TRIUMPH-4 study will assess the subset of obese or overweight participants with concomitant knee osteoarthritis, after being treated with retatrutide. It has an enrolment goal of 405 patients, and it is estimated to last until December 2025.

The planning and execution of additional phase III studies could offer in-depth insights into the critical factors influencing weight reduction. It is vital to understand that achieving weight loss is not always beneficial for health; in certain circumstances, it may lead to clinically significant declines in muscle and bone mass rather than a target reduction in fat. This phenomenon raises important questions about the quality and composition of weight loss, as retaining lean body mass is essential for maintaining overall strength, mobility, and metabolic health. Therefore, there is a need to design studies that not only focus on the percentage but also investigate the body composition changes that accompany weight loss [46,47,48,49].

It should be noted, however, that while retatrutide shows promise as a weight-loss therapy for adults, its use is not currently approved for children and adolescents. The safety and efficacy of retatrutide have not been established in pediatric populations. Current clinical trials have focused exclusively on adults, and there is a lack of data regarding its use in children and adolescents. Therefore, retatrutide is not indicated for individuals under 18 years of age. For children and adolescents, obesity prevention and management should prioritize non-pharmacological interventions. Lifestyle modifications, including a balanced diet and regular physical activity, are the cornerstone of pediatric obesity prevention. The U.S. Preventive Services Task Force recommends that clinicians screen for obesity in children and adolescents aged 6 years and older and offer or refer them to comprehensive, intensive behavioural interventions to promote improvements in weight status [50].

## 8. Conclusions

In conclusion, retatrutide appears to herald a significant advancement in the realm of pharmacotherapy for obesity and weight management (Figure 2). With its remarkable safety profile and noteworthy percentages of weight reduction, it undeniably generates considerable enthusiasm within the medical community. However, it is essential to emphasize that additional phase 3 clinical trials are crucial to further evaluate the drug’s safety and long-term effects.

Moreover, the financial implications of this groundbreaking medication cannot be ignored; despite its promising attributes, it may remain an inaccessible option for patients who cannot afford such treatments. Furthermore, studies assessing the quality of weight loss beyond simple BMI reduction are lacking, underscoring a significant area that warrants deeper research. This gap highlights the need for comprehensive studies that evaluate not only the effectiveness of weight loss but also its overall impact on patients’ health and well-being.

## Figures and Tables

**Figure 1 biomolecules-15-00796-f001:**
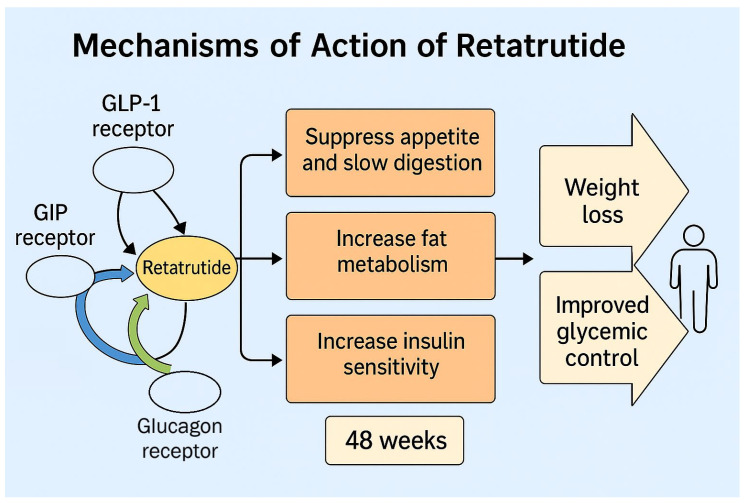
Retatrutide’s mechanisms of action.

**Figure 2 biomolecules-15-00796-f002:**
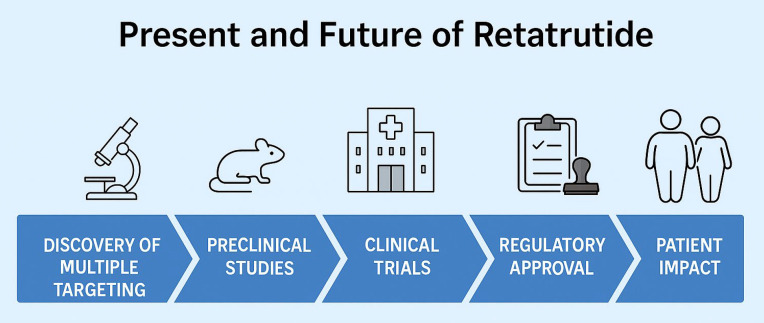
The present and future of Retatrutide.

**Table 1 biomolecules-15-00796-t001:** Concise summary of key retatrutide studies.

Type of Study	Year	Studies
Animal Study (mice)	2017	Jall et al.
Animal Study (mice)	2023	Urva et al.
Animal Study (mice)	2024	Ma et al.
Phase I Clinical Study	2022	Coskun et al.
Phase I Clinical Study	2022	Urva et al.
Phase II Clinical Study	2023	Jastreboff et al.
Phase II Clinical Study	2023	Rosenstock et al.
Phase II Clinical Study	2024	Sanyal et al.

## Data Availability

The original contributions presented in this study are included in the article. Further inquiries can be directed to the corresponding author.

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
