# Peer review of "Retatrutide—A Game Changer in Obesity Pharmacotherapy"

_biomolecules, 2025, doi:10.3390/biom15060796_

Round 1
Reviewer 1 Report
Comments and Suggestions for Authors
This review is seated with good idea but with low scientific significance and values since the authors describe the very known things inside. Also, the organization of this nonsystematic article is poor and must be much more focused on actual investigations and new scientific approach in this theme. Reference must me newly, 20 from 43 are older more than 5 years, so these citations provide un-relevanced article and information presented inside. Also, the number of cited articles must not be less than 50 in review article. This article does not fit the basic guidelines in writing medical review article according to the International Committee of Medical Journal. Editors (ICMJE) .
Comments on the Quality of English LanguageThis review is seated with good idea but with low scientific significance and values since the authors describe the very known things inside. Also, the organization of this nonsystematic article is poor and must be much more focused on actual investigations and new scientific approach in this theme. Reference must me newly, 20 from 43 are older more than 5 years, so these citations provide un-relevanced article and information presented inside. Also, the number of cited articles must not be less than 50 in review article. This article does not fit the basic guidelines in writing medical review article according to the International Committee of Medical Journal. Editors (ICMJE) .
Author Response
Please see the uploaded attachment.

Reviewer 2 Report
Comments and Suggestions for Authors
This manuscript reviews the preclinical and clinical data obtained with the triple incretin agonist, Retatrutide which is now in phase III (Triumph 4 study). The manuscript describes first the main effects observed with either GLP-1r, GIPr or Glucagon receptors single agonists. It presents afterwards the preclinical studies obtained mainly in rodents. The last paragraphs deal with phase I, II and ongoing phase III trials. The need for comprehensive body composition analysis is well described - an important problem associated with sarcopenia after long lasting usage of GLP-1RA.
Page 3, line 93. Hypothalamus must be changed into brain since focusing on the hypothalamus will be confusing especially looking at the following paragraph.
Page 3, line 113. Reference is needed about the possible reduction of gastric emptying after GIPr agonist.
Page 4, line 151 - relative pKa are needed to support the potencies (including the species for which these pKa were computed)
Page 4, line 159 - reference is mandatory about therapy discontinuation since it is a critical problem with GLP-1RA - up to 80% discontinuation after 5 years in obese patients, 57% after 2 years in real life studies.
Page 4, line 172 and following - Critical review about GE measurement used in reference 34 is mandatory since the method is not transposable to gold standard scintigraphic methods and does not involved tracers. Furthermore, the method used did not takes into account secretion based error.
Page 5 line 213 and following - Critical review of the paper 36 is also mandatory since GLP-1RA is absent in the liver. What is the rationale for a reversal of steatohepatitis.
Author Response
Please see the uploaded attachment.

Reviewer 3 Report
Comments and Suggestions for Authors
The manuscript (review) is very interesting. The authors present a very interesting and updated proposal regarding the effects of Retatrutide. In the review, the authors conducted a detailed analysis of the various studies conducted in humans (phase I and II), as well as the background information obtained from animal studies. Therefore, this review could be very useful to researchers and clinicians. However, I have the following comments.
I. Comments:
1. Improve the wording of the review's objective.
2. After the introduction, I suggest including a section (methodology) describing the criteria used to select the cited manuscripts.
3. I suggest replacing subheading 3. Introducing Retatrutide with "General Aspects of Retatrutide."
4. The authors provided a very good description of Retatrutide's mechanisms of action. However, Figure 1 is very general. In this regard, the authors could propose two figures. In Figure 1, present the main mechanisms that explain Retatrutide's effects, and in the second figure, show The Present and Future of Retatrutide (current Figure 1).
5. The obesity epidemic, T2DM, MAFLD, etc., have in common an unhealthy diet and a sedentary lifestyle. The authors could briefly address these aspects.
6. Unfortunately, obesity, insulin resistance, and fatty liver disease are showing constant and rapid growth in children and adolescents. This indicates that the public health problem will worsen in the coming years. In this context, it would be very good if the authors briefly reported that Retatrutide is indicated for adults, and in the case of children and adolescents, interventions should be preventive (diet and physical activity).
Author Response
Please see the uploaded attachment.

Round 2
Reviewer 1 Report
Comments and Suggestions for Authors
Authors are answered on all request and I recommend to accept this manuscript in present form.
Reviewer 3 Report
Comments and Suggestions for Authors
Authors answered all my comments and questions. Therefore, the manuscript can be accepted .